# Anatomical and functional maturation of the mid-gestation human enteric nervous system

Lori B. Dershowitz [1,2], Li Li[3], Anca M. Pasca [3] & Julia A. Kaltschmidt [1,2]

Immature gastrointestinal motility impedes preterm infant survival. The enteric nervous system controls gastrointestinal motility, yet it is unknown when the human enteric nervous system matures enough to carry out vital functions. Here we demonstrate that the second trimester human fetal enteric nervous system takes on a striped organization akin to the embryonic mouse. Further, we perform ex vivo functional assays of human fetal tissue and find that human fetal gastrointestinal motility matures in a similar progression to embryonic mouse gastrointestinal motility. Together, this provides critical knowledge, which facilitates comparisons with common animal models to advance translational disease investigations and testing of pharmacological agents to enhance gastrointestinal motility in prematurity.

Gastrointestinal (GI) immaturity is a major source of morbidity and mortality amongst extremely preterm infants born at less than 28 postconceptional weeks (PCW). The clinical care of these patients requires early initiation of enteral feeds despite immature GI motility[1]. Studies show that lack of enteral feeds leads to intestinal atrophy and increased mortality[2]; however, initiation of enteral feeds is risky in itself. Extremely preterm infants have high rates of feeding intolerance, functional intestinal obstruction, and overwhelming infections. Effective medical treatments for GI complications are lacking. Clinical trials on extremely preterm infants are highly challenging, and attempts to use adult prokinetic agents such as erythromycin have failed to treat feeding intolerance in the preterm population[3,4]. Therefore, a fundamental understanding of mid-gestation GI physiology is critical for developing therapies.

The enteric nervous system (ENS), an autonomous branch of the peripheral nervous system, resides within the intestinal walls and controls GI function. The ENS arises from migratory neural crest cells. In humans, these cells enter the developing intestine at 4PCW and have colonized the intestinal length by 7PCW, and this intrinsic innervation precedes extrinsic intestinal innervation, which is not present by 9PCW[5–7]. Whole-intestine single-cell RNA sequencing (scRNAseq) has revealed that enteric neurons are present in the intestine as early as 6PCW and that genes indicative of functionally distinct enteric neuron classes are expressed during the first trimester[8–10]. Visualization of the human fetal ENS using cross sections, which show the ENS within the intestinal walls, have identified structural changes with the emergence of individual ganglia evident at 14PCW[6,11,12]. Structural progression has also been observed in non-neuronal cells of the human ENS with interstitial cells of Cajal, pacemaker cells of the intestine, forming a more mesh-like appearance later in gestation[13].

The ENS can also be visualized in wholemount preparations, in which the ENS is viewed as a sheet. Using this approach in mice, we recently demonstrated a previously unknown organization of the ENS into circumferentially oriented stripes[14]. These neuronal stripes arise from gradual reorganization of enteric neurons during late embryonic and early postnatal ages[14], which corresponds to the emergence of neurogenic GI motility at embryonic day 18.5 in the mouse small intestine[15,16]. In mouse colon, the emergence of neuronal stripes and the onset of neurogenic GI motility both occur in the early postnatal period[14,15]. Whether the human ENS undergoes similar reorganization into stripes and whether a similar relationship between ENS structural and functional development is conserved in human is not understood.

[1]Department of Neurosurgery, Stanford University School of Medicine, Stanford, CA 94305, USA. [2]Wu Tsai Neurosciences Institute, Stanford University, Stanford, CA 94305, USA. [3]Department of Pediatrics, Stanford University School of Medicine, Stanford, CA 94305, USA. ✉e-mail: apasca@stanford.edu; jukalts@stanford.edu

Collectively, these mouse and human studies suggest that wholemount analysis of the human fetal ENS may reveal structural changes that inform GI function.

The physiology of extreme prematurity is comparable to late second trimester fetal development, yet histological and scRNAseq studies of the human fetal ENS primarily examine first trimester tissue[6–9]. Similarly, GI motility assessments have been restricted to manometry studies in preterm and term infants from 28–42PCW. These recordings primarily identified random and non-propagating GI activity at preterm ages and did not detect migrating motor complexes, a form of neuronally mediated GI motility, until 37PCW[17,18]. Here, we assessed human fetal ENS development in wholemount preparations from second trimester development, between 13–23PCW. Further, we related these structural findings to development of human fetal GI motility ex vivo, the first functional analyses of ex vivo human fetal tissues. This knowledge of ENS development from a period that closely precedes extremely preterm birth is essential for understanding the pathophysiology of severe GI disorders and for guiding new enteral feeding regimens and pharmacologic interventions for GI immaturity.

## Results

The intestinal wall contains five distinct layers: mucosa, submucosal plexus (SMP) of the ENS, circular muscle (CM), myenteric plexus (MP) of the ENS, and longitudinal muscle (LM) (Fig. 1b). To identify morphological stages of ENS development in the second trimester of pregnancy, we collected fresh ex vivo human intestines from nondiseased fetuses between 13–23PCW (Fig. 1a). For analyses, we subdivided the intestine into five anatomically and functionally distinct regions (duodenum, jejunum, ileum, proximal colon, and distal colon) (Fig. 1a). Immunolabeling of jejunum and proximal colon with panneuronal markers PGP9.5 and HuC/D revealed a uniform MP with few neurons in the SMP at 14PCW (Fig. 1c, Supplementary Fig. 1a). By 23PCW, enteric neurons have condensed into discrete ganglia in both the SMP and MP, and the CM and mucosa contain a greater density of neuronal fibers, which may arise from either extrinsic or intrinsic innervation (Fig. 1c, Supplementary Fig. 1a). We also observed morphological changes in other intestinal tissues, although this part of the analysis was limited by sample number and potential variability in tissue embedding. The mucosa becomes more elaborate from 14 to 23PCW (Fig. 1a, Supplementary Fig. 1a). Analysis of smooth muscle thickness from cross sections demonstrated that thickness of the smooth muscle layers triples in the colon while remaining stable in the small intestine (Supplementary Fig. 1f, g). Labeling the MP in cross section with Sox10, which marks non-neuronal ENS cells including precursors and glia, revealed that the density of Sox10+ cells decreases over gestational time in the small intestine MP but is unchanged in the colon (Supplementary Fig. 1h,i). Thus, human intestinal and ENS morphology undergo significant maturation during mid-gestation with regional differences.

In mouse, the embryonic and neonatal MP undergoes progressive reorganization from a random array into circumferentially oriented neuronal stripes, which define the locations of enteric ganglia[14]. To determine whether the human fetal ENS undergoes similar structural reorganization, we visualized enteric neurons in MP wholemounts at ages spanning 14–23PCW (Fig. 1b, d). In the duodenum MP, enteric neurons reside in circumferentially oriented neuronal stripes as early as 14PCW (Fig. 1d). In the distal colon MP, distinct neuronal stripes are not evident until 21PCW (Fig. 1d). The emergence of neuronal stripes is not due to programmed cell death as labeling with apoptotic marker Caspase 3 in both wholemount and cross section preparations identified no apoptotic neurons at 14PCW, and less than 0.6% of HuC/D+ neurons express Caspase 3 at 23PCW (Supplementary Fig. 1c–e). To validate these observations with a computational non-biased approach, we generated spatial probability maps of neuronal

location within our tissue (Fig. 1f). These maps clearly identify neuronal stripes in the duodenum at 18PCW and in the colon at 23PCW (Fig. 1f). z-score comparisons between neuronal data and synthetically generated pseudorandom data further demonstrated that neuronal organization becomes non-random and more clustered in the small intestine prior to the colon (Fig. 1d–g)[14]. Additionally, this analysis identified neurons within either of two distinct positions: (1) neurons within stripes, which reside close to the CM and (2) bridging neurons, which reside closer to the LM (Fig. 1d–f). We primarily focused on MP development due to its role in GI motility, but we also noted structural changes in SMP wholemounts including increased ganglia size from 15 to 23PCW (Supplementary Fig. 1b). Taken together, these results demonstrate that enteric neurons in the mid-gestation fetal MP undergo substantial reorganization, which occurs in the small intestine prior to the colon. The reorganization of neuron cell bodies generated stripes with an overlying grid-like structure, an organization reminiscent of, but more elaborate than, neuronal stripes in mouse[14]. This observation of enteric neuron cell body organization complements the known lattice structure of enteric neuronal processes, which has been described in the embryonic mouse, fetal human, and adult human ENS[19–21].

The basic GI motility circuit in rodents contains four functionally distinct neuron classes[22]. scRNAseq studies of the mouse ENS have identified putative markers of enteric neuronal subtypes[22–24], and we have recently identified subsets of these markers that reliably label neuronal cell bodies from different enteric neuron classes in the adult mouse ENS[14]. Using these prior studies as a basis, we next assessed the quantity and distribution of the main enteric neuron classes across human MP development. We labeled MP wholemounts with antibodies against calretinin (CalR), neuronal nitric oxide synthase (nNOS), neurofilament-M (NF-M), and somatostatin (SST), which broadly define subsets of excitatory motor neurons, inhibitory motor neurons, sensory neurons, and interneurons in mice, respectively (Fig. 1h–k, Supplementary Fig. 1j–m)[22–24]. These markers approximate four neuronal classes, though CalR expression has also been observed in sensory neurons, and NF-M expression has been noted in some motor neurons[23,25,26]. The percentage of CalR+HuC/D+ neurons increased over gestational time in all intestinal regions while the percentage of nNOS+HuC/D+ neurons appeared mostly stable with a slight decrease over time in the duodenum and jejunum (Fig. 1h–k). Labeling with NF-M did not reveal an obvious trend across gestational time or region, and SST+HuC/D+ neurons remained scarce until 21PCW (Supplementary Fig. 1j–m). Thus, the mid-gestation human MP contains distinct neuron classes with apparent increasing representation of CalR+ subtypes and decreasing representation of nNOS+ subtypes.

Finally, we assessed how these organizational changes relate to human GI motility development. We established a GI motility monitor used in guinea pig and mouse[27–29] for ex vivo motility analysis of the second trimester human small intestine (Supplementary movies 1–8). Using spatiotemporal maps (STMs), which capture changes in intestinal diameter over time, we identified contractions that travel both proximally and distally from initiation points at 13–15PCW (Fig. 2a, b, Supplementary Fig. 2a, and Supplementary movie 1). At 18PCW, we still observed these bidirectional contractions, but they occur intermittently in time with intervening periods of quiescence (Fig. 2a, Supplementary movie 2). Notably, at 21–22PCW we ceased to observe bidirectional events and instead found evidence of contractions that propagate exclusively from proximal to distal (Fig. 2a, b, Supplementary Fig. 2a, b, and Supplementary movies 3, 4). To determine whether these GI motility patterns are primarily myogenic ripples[16] or neuronally mediated contractions, we inhibited neuronal activity using tetrodotoxin (TTX), a non-specific sodium channel blocker. At 13–15PCW, TTX treatment affected contraction frequency in some individual samples but did not alter the overall shape of GI motility patterns (Fig. 2c, d and Supplementary movies 5, 6). Therefore, these

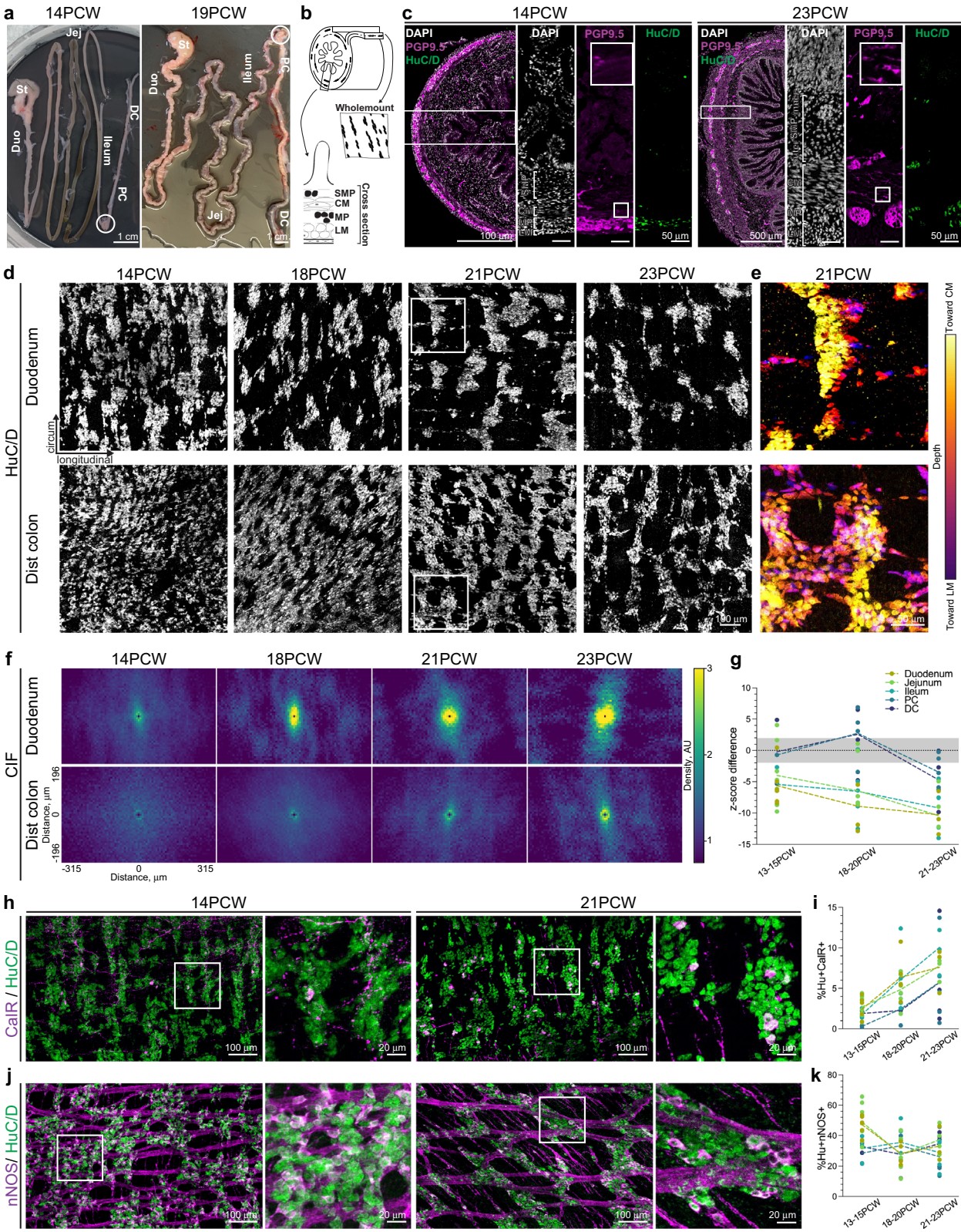

patterns are putative myogenic ripples that have been described in mouse[15,16]. In all samples examined from 18–22PCW, TTX treatment dramatically affected GI motility patterns (Fig. 2c, Supplementary Fig. 2b, and Supplementary movies 7–10). At 18PCW, TTX treatment ablated quiescent periods and resulted in constant bidirectional motility events (Fig. 2c and Supplementary movies 7, 8). At 21–22PCW, TTX substantially altered GI motility and disrupted the exclusively distal

direction of contractions (Fig. 2c, Supplementary Fig. 2b, and Supplementary movies 9, 10). Samples from 21–22PCW exhibit few myogenic ripples at baseline and increased myogenic ripple frequency with TTX treatment (Fig. 2d). Collectively, these results suggest that GI motility is limited to myogenic ripples at 13–15PCW and that the ENS may first influence GI motility beginning at 18PCW with distally propagating contractions evident by 21–22PCW.

**Fig. 1 | Anatomical development of the mid-gestation human intestine and ENS.**
**a** Images of 14 and 19PCW human fetal intestinal samples with regional subdivisions indicated. White circles denote the appendix. **b** Schematic of the intestinal tube with the two ENS plexuses indicated by black ovals. The ENS can be viewed in cross section (left) or as a wholemount (right). **c** Representative images of cross sections from the proximal colon at 14 and 23PCW with immunohistochemical (IHC) labeling against DAPI (white), pan-neuronal markers PGP9.5 (magenta) and HuC/D (green). White boxes indicate locations of higher magnitude insets. Brackets denote location of SMP, CM, MP, and LM. *n* = 2 14PCW and 3 23PCW.
**d, f, g** Representative images of HuC/D labeling in MP wholemounts of the duodenum (top) and distal colon (bottom) over gestational time are shown in (**d**). Scale bar applies to all images. Conditional intensity function (CIF) plots in (**f**) are derived from HuC/D labeling in MP wholemounts of the duodenum (top) and distal colon (bottom) over gestational time. Yellow: high probability density; blue: low. Axes apply to all panels. Accompanying z-score difference (**g**) between data and mean of synthetically generated pseudorandom values in all intestinal regions over gestational time. Gray bar indicates random value range. *n* = 1 PC 13–15PCW; *n* = 2 DC 13–15PCW; *n* = 3 duo 21–23PCW and ileum 13–15PCW; *n* = 4 duo 18–20PCW, jej 21–23PCW, PC 18–20PCW, and DC 18–20PCW; *n* = 5 ileum 18–20PCW, ileum 21–23PCW, PC 21–23PCW, and DC 21–23PCW; *n* = 6 duo 13–15PCW and jej

18–20PCW; and *n* = 8 jej 13–15PCW. Trendlines interconnect means across age groups. (**e**) Depth projection of HuC/D+ neurons in the 21PCW MP as indicated by white boxes in (**d**). Yellow: closer to CM; purple: closer to LM. Scale bar applies to all images. **h–k** Representative images of HuC/D (green) and either excitatory marker CalR (**h**, purple) or inhibitory marker nNOS (**j**, purple) in the jejunum MP at 14 and 21PCW. White boxes indicate locations of higher magnitude insets. Proportion of total HuC/D+ neurons positive for CalR (**i**) or nNOS (**k**) over gestational time. For CalR, *n* = 1 DC 18–20PCW; *n* = 2 PC 13–15PCW and DC 13–15PCW; *n* = 3 duo 21–23PCW, PC 18–20PCW, and PC 21–23PCW; *n* = 4 jej 21–23PCW, ileum 13–15PCW, ileum 18–20PCW, and DC 21–23PCW; *n* = 5 duo 18–20PCW and ileum 21–23PCW; *n* = 6 duo 13–15PCW and jej 18–20PCW; and *n* = 7 jej 13–15PCW. For nNOS, *n* = 2 PC 13–15PCW, DC 13–15PCW, and DC 18–20PCW; *n* = 3 duo 21–23PCW, ileum 13–15PCW, and PC 18–20PCW; *n* = 4 duo 18–20PCW, jej 21–23PCW, ileum 18–20PCW, PC 21–23PCW, and DC 21–23PCW; *n* = 5 jej 18–20PCW and ileum 21–23PCW; *n* = 6 duo 13–15PCW; and *n* = 8 jej 13–15PCW. Trendlines interconnect means across age groups. Scale bars as indicated. Intestinal regions are noted with distinct colors as indicated in (**g**). AU arbitrary units, CM circular muscle, Duo duodenum, DC distal colon, Jej jejunum, LM longitudinal muscle, MP myenteric plexus, PC proximal colon, PCW postconceptional week, SMP submucosal plexus. Source data are provided as a Source Data file.

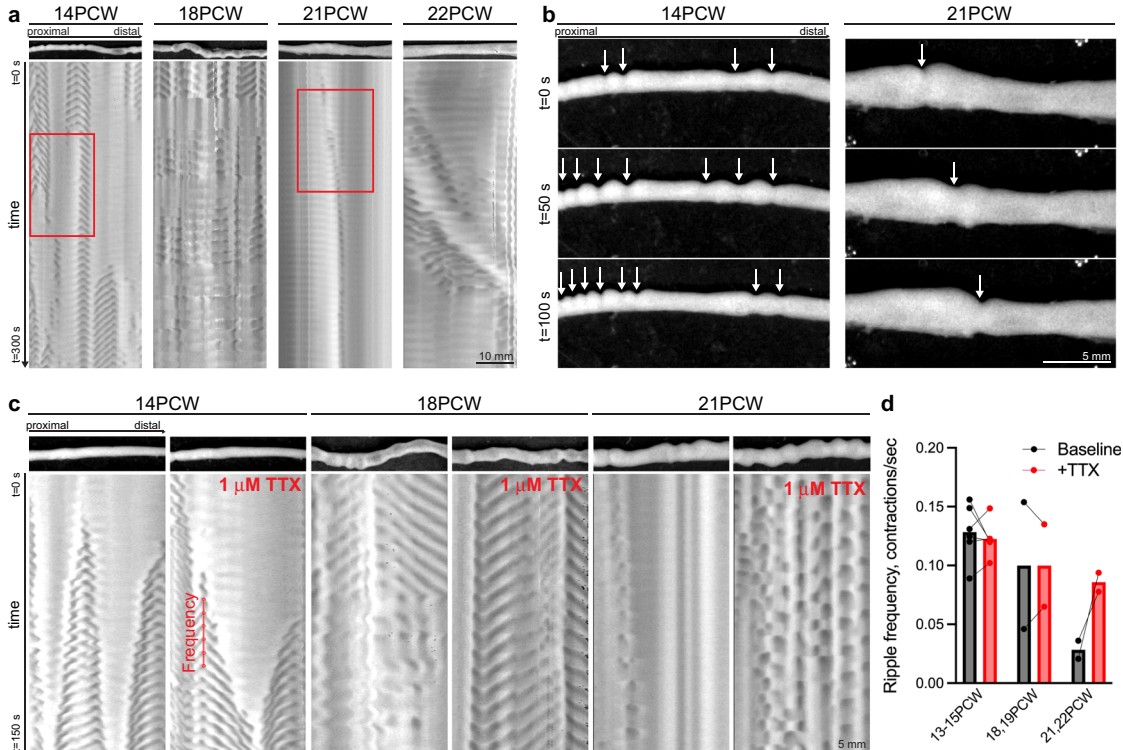

**Fig. 2 | GI motility in the second trimester human small intestine assessed ex vivo. a** Spatiotemporal maps (STMs) with accompanying images of GI motility in the human fetal jejunum at 14, 18, 21, and 22PCW at baseline (left panels) and with the addition of 1 μM tetrodotoxin diameter; light gray: increased. STMs are a subset of those in Supplementary Figure 2a. **b** Images from video recordings over three timepoints indicated by red boxes in (**a**) from 14 and 21PCW. Arrows indicate locations of contractions. **c** STMs

with representative images of GI motility in the human fetal duodenum at 14, 18, and 21PCW at baseline (left panels) and with the addition of 1 μM tetrodotoxin (TTX, right panels). Dark gray: decreased intestinal diameter; light gray: increased. Scale bar applies to all images. **d** Frequency of contractions as indicated in (c) at baseline (black bars) and with TTX (red bars). *n* = 2 from 21, 22PCW and 18, 19PCW, and *n* = 5 from 13–15PCW. Source data are provided as a Source Data file.

## Discussion

Here, we characterized the anatomical and functional development of the human fetal ENS with a focus on second trimester in utero development. First, we identified emerging striped neuronal cytoarchitecture of the human MP, which arises earlier in the small intestine than in the colon (Fig. 1d–g), a result similar to mouse MP development at late embryonic and early neonatal ages[14]. Interestingly, we also observed a neuronal grid overlying the stripes (Fig. 1d–f), a more complex macrostructure than observed in mouse[14]. Understanding

this structural progression is of major clinical significance for neonatology and maternal-fetal medicine because this process may be disrupted in cases of utero-placental insufficiency, which results in greater reduction of fetal intestine growth as compared to growth of other organs[30].

Second, our findings suggest that the representation of CalR+ excitatory neurons increases relative to nNOS+ inhibitory neurons in the human MP during the second trimester (Fig. 1h–k). This result is an important clinical benchmark because a decrease in the ratio of

CalR+/nNOS+ enteric neurons has been observed in the ganglionated portion of the intestine of mouse models and individuals with Hirschsprung's disease[31].

Third, we establish the first ex vivo assay of function in an intact human fetal organ. Our ex vivo GI motility analysis identified myogenic ripples at 13-15PCW, which are known to be the earliest form of GI motility in the mouse embryonic intestine (Fig. 2a–c, Supplementary Fig. 2a, and Supplementary movies 1, 5, 6). We also found a gradual progression of neurogenic GI motility from intermittent active/quiescent periods at 18PCW to distally propagating contractions at 21–22PCW (Fig. 2a–c Supplementary Fig. 2a, b, and Supplementary movies 2–4, 7–10). The timeline of GI motility maturation from myogenic ripples to neurogenic motility mirrors development of GI motility in the mouse embryonic small intestine, which exhibits myogenic ripples at mid-gestation and neurogenic GI motility just prior to birth[15,16]. While future studies will need to include more gestational ages, our findings suggest incipient maturation of the ENS motility circuit beginning at 18PCW with more mature patterns at 21-22PCW, ages potentially amenable to clinical interventions to enhance ENS function in preterm infants. Collectively, our studies provide unprecedented insights into the mid-gestation human ENS that allow for direct comparisons between human and mouse ENS development and may improve use of animal models for studying GI diseases with an in utero onset. Further, we demonstrated that the ex vivo platform can be used with pharmacologic agents thus enabling noninvasive screening of drugs that enhance GI motility with the long-term goal of reducing feeding intolerance and serious complications including intestinal perforation and sepsis.

## Methods

### Ethics and inclusion statement

All authors met the authorship inclusion criteria for Nature Portfolio journals. Research was conducted at Stanford University, where all authors are affiliated. The role of each author was discussed and agreed upon prior to completion of the project.

Approval for fetal tissue collection for research was obtained from the Institutional Review Board at Stanford University. All tissue samples for this study were collected under an approved protocol through the Research Compliance Office at Stanford University, which ensures research compliance with policies both within and outside of Stanford University. All women scheduled to have a pregnancy termination at Stanford had the opportunity to participate by providing written consent to donate fetal tissue for research. All women who consented to donate tissue were eligible to participate in our study. No monetary compensation was provided for consent to use fetal tissue as donation for research. All tissues collected were negative for trisomies, based on prenatal screening. To protect privacy, the age of the woman and the sex of the fetus were not made available to the research team. Therefore, sex was not considered in this study. The Stanford Institute Tissue Donation Policy does not require specific consent for the use of donated tissues in experiments that do not involve cell culture or the creation of cell lines.

### Human tissue collection

De-identified intestinal tissue samples were obtained after elective pregnancy termination at Stanford University School of Medicine under an approved protocol through the Research Compliance Office at Stanford University. Tissue postconceptional week (PCW) was estimated based on ultrasound appearance and last menstrual period prior to pregnancy termination (Supplementary Table 1). For analysis, PCW ages were rounded to the closest week. For histology experiments, all intestinal tissue was immediately placed in ice-cold PBS. For functional experiments, tissue was immediately placed in carbogenated (95% $CO_2$, 5% $O_2$) Krebs solution (pH 7.4 containing (in mmol/l): 117 NaCl, 4.7 KCl, 3.3

$CaCl_2$ ($2H_2O$), 1.5 $MgCl_2$ ($6H_2O$), 25 $NaHCO_3$, 1.2 $NaH_2PO_4$ and 11 Glucose) at 37 °C. Given differences in quality of tissue, histological analyses were only included if individual neurons could be visually discerned.

### Immunohistochemistry

For histological assessment of intestinal and enteric nervous system (ENS) structure, intestinal tissue was identified and dissected from surrounding organs, and the mesentery cut away. Based on tissue coloration and position relative to anatomical landmarks including the stomach and appendix (Fig. 1a), the intestines were subdivided into stomach, duodenum, jejunum, ileum, proximal colon, and distal colon. For each region, a representative piece of tissue was collected. The length of each tissue piece varied by age, from ~4 cm in length at 14PCW and ~10 cm in length at 23PCW.

For cross sections, tissue was fixed as a tube with each end pinned on Sylgard 170 in 4% PFA for 2 h at 4 °C while shaking. Tissue was placed in 30% sucrose overnight, embedded in OCT, and stored at −80 °C. Tissue sections were cut at 12 mm thickness with a Leica cryostat CM3050 S. After being allowed to dry for 30 min, slides were rehydrated with 3 washes in ice-cold PBS and placed in blocking solution with 0.5% donkey serum for 1 h at room temperature. Slides were processed with a Streptavidin/Biotin Blocking Kit (Vector Laboratories) per package instructions and incubated in primary antibodies (see Supplementary Table 2) diluted in PBT (PBS, 1% BSA, 0.1% Triton X-100) overnight at 4 °C. Slides were washed 3 times in ice-cold PBS, incubated in PBT containing secondary antibodies (see Supplementary Table 3) for 2 h at room temperature, and washed 3 times in ice-cold PBS. Slides were rinsed in ddH2O and coverslipped with Fluoromount-G (Southern Biotech).

For wholemount preparations, each tissue piece was cut open along the mesenteric border and tautly pinned mucosa-side up on Sylgard 170 in ice-cold PBS. A paintbrush was used to gently remove any intestinal contents. Full-thickness tissue preparations were fixed in 4% PFA for 2 h at 4 °C while shaking. After fixation and 3 washes in ice-cold PBS, tissue was pinned mucosa-side down on Sylgard 170, and fine forceps were used to peel the muscularis (smooth muscle layers and myenteric plexus (MP)) away from the underlying mucosa. The mucosa and muscularis layers were either used immediately or stored in PBS with 0.1% sodium azide at 4 °C for up to 1 month. For immunostaining, tissue was cut into ~2 cm long pieces and placed into WHO microtitration trays (International Scientific Supplies) with PBT. Tissue was then placed in primary antibodies (see Supplementary Table 2) diluted in PBT overnight at 4 °C while shaking. Tissue was washed 3 times in PBT, incubated in PBT containing secondary antibodies (see Supplementary Table 3) for 2 h at room temperature while shaking, washed twice in PBT, treated with PBS containing DAPI (1 mg/mL) for 5 min, and washed twice in PBS. For mounting, tissue was submerged in ddH2O, gently placed onto slides using paintbrushes under a dissection microscope, and allowed to briefly air dry before being coverslipped with Fluoromount-G (Southern Biotech).

### Image acquisition

Images were captured on a Leica SP8 confocal microscope using a 20× (NA 0.75) oil objective. Acquisition areas were selected, acquired, and stitched with LASX Navigator Mode (Leica). Images were captured as z-stacks, which were set at 3 μm intervals capturing the full depth of the area of interest, such as the MP. Images were subsequently processed with ImageJ/FIJI (NIH, Bethesda, MD) and Photoshop (Adobe).

### Image analysis

**Depth projection.** All image analyses were performed in ImageJ/FIJI (NIH, Bethesda, MD). Depth projection images of HuC/D labeling in the MP were generated and pseudocolored from z-stacks using the Temporal-Color Code macro in ImageJ/FIJI.

**Conditional intensity function (CIF) analysis.** Spatial probability maps were generated with CIF analysis as previously described[14] with some modifications. Briefly, 800 × 800 μm maximum intensity projections of HuC/D+ neurons in MP wholemounts were processed with Gaussian blur, Threshold, and Watershed functions. Analyze Particles was used to determine the XY coordinates of neurons in the tissue. Results were further analyzed with CIF algorithm, which assesses the number of neurons in a 2D grid around each neuron in the tissue. Results were smoothed with a Gaussian of 20 μm standard deviation. All analyses were performed in Python.

**z-score difference analysis.** Comparison of z-scores between neuronal data and synthetic data was performed as previously described[14] with some modifications. Briefly, to determine the deviation of neuronal organization from complete spatial randomness over developmental time, XY coordinates of HuC/D+ neurons were determined with Analyze Particles as above and assigned a z-score that reflects the deviation from the value expected with randomness. Five hundred samples of synthetic data were generated, and the mean z-score of these trials was assessed. In the generation of synthetic pseudorandom data, a minimum distance of 5 μm was imposed because the neuronal data did not overlap in space. To calculate the z-score difference, the z-score from 500 random samples was subtracted from the z-score of the data. All analyses were performed in Python.

**Neuronal subtype counting.** To count neuronal subtypes as an overall percentage of HuC/D+ neurons, image stacks of HuC/D immunolabeling in MP wholemount preparations were blurred and thresholded. Because HuC/D+ neurons in our tissue were tightly packed and not easily resolved, we approximated neuronal number from the overall area of HuC/D labeling. The HuC/D stack was maximally projected, and the Analyze Particles function was used to determine the total area labeled with HuC/D. To estimate the average area of a human fetal HuC/D+ enteric neuron, neuronal diameter was measured for 10 HuC/D+ neurons from multiple regions per sample. The average neuronal diameter was similar regardless of gestational age, and the average across ages was 9.374 μm (Supplementary Fig. 3). From this value, neuronal area was approximated as 138 μm². Total HuC/D area was divided by this value to approximate the number of neurons in an image. To determine the percentage of HuC/D+ neurons that express a given subtype marker, stacks labeled with a subtype marker were blurred and combined with the processed HuC/D stack using the Image Calculator function. This result was maximally projected, and the number of cells expressing HuC/D and a given subtype marker was counted with Analyze Particles. This value was divided by the approximate number of HuC/D+ neurons to determine the overall representation of the subtype. To quantify HuC/D+ neurons expressing apoptotic marker Caspase 3, the same approach was taken as above except cryosections of intestinal cross sections were utilized. The polygon selection tool was used to select and isolate only the MP. Two sections were analyzed per sample.

**Muscle thickness quantification.** Cyrosections of intestinal cross sections labeled with smooth muscle marker SMA we imported into Fiji. Thickness was measured with the line segment and measure tools. Per sample, the thickness of both the circular and longitudinal muscle was measured at two locations in each cryosection, and two cryosections were analyzed per sample.

**Sox10 density quantification.** Cryosections of intestinal cross sections labeled with Sox10 were imported into Fiji. A segment of MP with an area of approximately 100,000 μm² was analyzed per section, and two sections were analyzed per sample. The number of Sox10+ cells in a given region was analyzed as above using the Gaussian blur, Threshold, and Analyze Particles functions in Fiji.

## Ex vivo gastrointestinal (GI) motility assay and analysis

**Ex vivo GI motility monitor.** GI motility was assessed ex vivo using a setup adapted from platforms previously described[27–29]. Tissue was placed in a dissection dish with Sylgard 170 and submerged in carbogenated Krebs solution as detailed above at 37 °C. The mesentery was cut away, and the tissue was moved to an organ bath above a heated water bath, which maintained the circulating carbogenated Krebs solution at 37 °C. The tissue was tautly pinned at the mesenteric border to the Sylgard 170 in the base of the chamber. After 10 min of acclimation, 10-min videos were recorded at 3.75 frames/s using IC capture software (Imaging Source) and a high-resolution monochromatic firewire industrial camera (Imaging Source®, DMK41AF02) connected to a 2/3″ 16 mm f/1.4 C-Mount Fixed Focal Lens (Fujinon HF16SA1) mounted above the organ bath. 30 min were captured at baseline followed by 30 min with the addition of 1 μM tetrodotoxin (TTX) (Alomone labs) to the circulating Krebs. A final 30 min of video were captured as the TTX solution was washed out. After completion of the experiment, tissue was collected, placed in ice-cold PBS, and processed for histology as described above. When the MP was removed, regions that did not contain a fully intact MP around their circumference were noted, and these regions were not subsequently analyzed for GI motility.

**Spatiotemporal map (STM) analysis.** STMs were generated from video recordings using VolumetryG9a software as previously described[27]. For each region analyzed, two videos per condition (baseline, TTX, and washout) were assessed. STMs were visually inspected for motility patterns and evidence of distally propagating contractions. Frequency of myogenic contractions, as indicated on the STM in Fig. 2c, was manually measured with the line segment and measurement functions in ImageJ/FIJI for a 1 min stretch of every STM containing myogenic ripples.

### Graphing and statistical analysis

All graphs and statistical analyses were generated with Prism 9 software (GraphPad). Each measurement represents a distinct human tissue sample. Given the low sample number for any given intestinal region and age, statistical outcomes were not reported in figures. Results of one-sided Tukey's correction for multiple comparisons can be found in Supplementary Table 4.

### Reporting summary

Further information on research design is available in the Nature Portfolio Reporting Summary linked to this article.

## Data availability

The data generated in this study are provided in Supplementary Information and Source Data files. Video data generated in this study are provided in Supplementary Video files. Step-by-step protocols are available upon request. Correspondence and requests for all other material should be addressed to L.B.D. or J.A.K. Source data are provided with this paper.

## Code availability

Code used to analyze these data is freely available on Github [https://github.com/druckmann-lab/EntericNervousSystemAnalysis.][14]

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

## Acknowledgements

We thank Kaltschmidt lab members for experimental advice and discussions of results. In particular, we thank Julieta Gomez-Frittelli for feedback on the manuscript. We thank Shaul Druckmann and Aiden Wang for sharing analysis software. We thank Subhamoy Das and Estelle Spear for setting up the ex vivo motility system in the Kaltschmidt lab. We are grateful to Grant W. Hennig for experimental advice and providing the Volumetry Software. This work was supported by Stanford Medical Scientist Training Program grants T32 GM007365-44 and T32-GM145402 (L.B.D.), a Stanford Maternal and Child Health Research Institute Pilot Grant postdoctoral fellowship (L.L.), a Stanford Maternal and Child Health Research Institute Pilot Grant, the Wu Tsai Neurosciences Institute, the Stanford University Department of Neurosurgery, and a research grant from The Firmenich Foundation (J.A.K.).

## Author contributions

L.B.D. designed, performed, and analyzed all experiments and co-wrote the manuscript. L.L. collected tissue samples. A.M.P. organized tissue collection, contributed to discussions about experimental design and data analyses, and edited the manuscript. J.A.K. designed and supervised experiments and co-wrote the manuscript.

## Competing interests

The authors declare no competing interests.
