## [Peer Review File · Nature Communications]

Anatomical and functional maturation of the mid-gestation human enteric nervous systemREVIEWER COMMENTS

Reviewer #1 (Remarks to the Author):

This manuscript describes research that fills a major hole in our knowledge of the embryonic development of the human gastrointestinal tract and in particular of the gut's intrinsic (enteric) nervous system. As such it will provide essential knowledge for both clinical and basic scientists interested in enteric neural function in the period immediately prior to "extremely preterm" birth and its parallels to late gestational development in mice, the ubiquitous animal model.

The manuscript is clear and the data-set is invaluable, so my comments below relate to ways the paper can be improved.

Line 33 "an" rather than "the"

Line 64 The increased density of nerve fibres referred to here is not obvious from the micrographs shown in Supplementary Fig 1A. The muscle layer is clearly thicker (although this could be made clearer by delineating the edges in some way), but density changes need quantification or higher magnification micrographs.

Line 84 The work of Chevalier et al 2021 (Communications Biology) describing an equivalent grid in mouse ENS development is relevant here

Line 91-94 This sentence over-simplifies the problem. Why was SOM used as a marker for interneurons?

Line 132 what does "de-prioritization" mean? Delayed or failure of?

Line 135-136 depends on whether calretinin is a universal or even a majority marker for excitatory motor neurons. What is the evidence about calretinin in human intrinsic sensory neurons or interneurons for example? Are there other excitatory motor neurons that are not calretinin?

Line 142-143 The ripples described here are remarkably similar to those identified in prenatal and immediately postnatal mouse colon by Roberts et al 2007 and 2010. This seems an important parallel, especially as this is the last period during which both the human preterm infant and the developing mouse gut are both free of an internal microbiome with its clear effects on enteric neuronal maturation (Collins et al 2013, Hung et al 2019, Poon et al 2020).

Reviewer #2 (Remarks to the Author):

In this manuscript, Dershowitz and colleagues use immunohistochemical staining and ex vivo motility assays to describe the neuroanatomy and contractility of the human fetal enteric nervous system (ENS) at 14-23 weeks post-conception (PCW). They describe their work as the "first description of human fetal ENS structural development" that is "foundational knowledge". Their major conclusions are that the ENS from mid-gestation fetuses undergo substantial reorganization, and that within the myenteric plexus (MP) there are distinct neuronal subtypes and representation of excitatory vs. inhibitory motor neuron populations that change over time. They found that starting at 21 PCW, contractions only propagated from proximal to distal. Using tetrodotoxin to block neuronal inputs, they found the ENS may first influence GI motility starting at 21PCW. The strengths of this manuscript are that it is clearly written, its focus on the human ENS, and examination of human fetal intestinal contractility. Many major deficits including the failure to acknowledge/cite previous work in this area, lack of experimental replicates, and superficial observations/characterization of the tissue, however, significantly diminish enthusiasm. These methodological issues limit what, if any, conclusions can be drawn from this work.

Major concerns:

1 – Previous groups have use IHC to describe enteric neurons and glia in relation to surrounding cells in the smooth muscle of the human fetal bowel, including in the second trimester. None of these studies were cited by the authors (PMID: 15672264, 32598482, 14991401, 12950074). These studies should be appropriately described and acknowledged to provide appropriate context for the current study, and also compel revision of the statement that the study under review is the “first description of human fetal ENS structural development”.

2 – Almost all of the data presented in Figures 1 and 2, and Supp Fig 1 have sample sizes of “1-3”. Human tissue is understandably difficult to acquire but showing data from a single subject and suggesting that it is “foundational knowledge” is inappropriate. Studies need to be powered for reproducibility. No reasonable biological conclusions can be drawn from observing motility in a single tissue sample, other than that contractions can be visualized ex vivo in the fetal intestine. Antenatal imaging already shows this to occur in vivo.

Furthermore, the way the data is presented in Figures 1f, 1i, 1j suggests that there are many samples from each PCW but the figure legend is not consistent. If there are multiple technical replicates from each biological replicate, then this should be clearly described as such.

3 – There are no data provided on the human subjects – inclusion/exclusion criteria, genetic defects, sex.

4 – Lack of characterization of tissue: Conclusions are drawn about myogenic vs. ENS mediated contractions based on studying a single subject yet there is no characterization of the cells of the smooth muscle syncytium at these ages. No examination of enteric glia. Similarly, they use calretinin and nNOS as markers of excitatory and inhibitory neurons, respectively, and draw conclusions about excitatory/inhibitory balance. However, it is not clear to what extent these designations are appropriate for the human fetal bowel. What is the evidence for this? Finally, SST is a neuropeptide and IHC for most neuropeptides in the adult ENS does not identify cell soma well because the neuropeptides are trafficked to the processes. Thus, it is not clear that the quantification in Supp 1g. is meaningful. Supp Fig 1e clearly shows SST immunoreactivity in fibers.

5 – There proximal to distal developmental gradient in the intestine is a well established concept for the ENS as well as many of the other key features of gut anatomy (epithelium, etc). Similarly, the stereotypic lattice-like structure of the ENS has been well described in numerous animal models and the human ENS over the course of many decades. Yet the authors re-brand this well-established organization as “neuronal stripes” and cite only their own study in mice as novel findings.

6 - Cell death characterization is weak. This should either be quantified/reproduced or removed.

7 – It is not clear what the analysis is in Fig 1e or what controls are done to illustrate that this is a meaningful assay for this tissue.

Minor:

Graphs in 2D/E seem to be missing 21 PCW, but text focuses on a change at 21PCW.

Reviewer #3 (Remarks to the Author):

This is an interesting and necessary study by Dershowitz et al on the anatomical and functionality of mid gestation intestine. Some suggestions to improve the manuscript are provided below:

Lines 65-68 to summarize Figure 1 saying “develop substantially” in morphology are vague and descriptive and should be described in more detail. The same feedback for lines 87-88 (“substantial reorganization”).

For Figure 1C, 1D, 1E it is unclear what the pictures are showing with regard to the anatomic orientation of the intestine. An orientation picture would be helpful.

For the data and results described in figure 2, how can the authors conclude that the ENS may first influence GI motility at 21 weeks without data from additional timepoints such as 15-20 weeks. In terms of the variability seen in humans, I would recommend these experiments be repeated with n= 3-4 per group at least, as an n of 1 is unacceptable to draw any conclusions.

Minor comments for improvement are below:

1. Minimize the use of acronyms throughout as it makes it difficult to read the manuscript. Acronyms when used do need to be spelled out the first time.
2. Videos didn't download correctly for this reviewer and so were unable to be viewed. Excessive use of videos is a weakness of the manuscript.

REVIEWER COMMENTS:

Reviewer #1:

This manuscript describes research that fills a major hole in our knowledge of the embryonic development of the human gastrointestinal tract and in particular of the gut's intrinsic (enteric) nervous system. As such it will provide essential knowledge for both clinical and basic scientists interested in enteric neural function in the period immediately prior to "extremely preterm" birth and its parallels to late gestational development in mice, the ubiquitous animal model. The manuscript is clear and the data-set is invaluable, so my comments below relate to ways the paper can be improved.

We greatly appreciate the overall enthusiasm of Reviewer #1. We also appreciate the thoughtful suggestions and have addressed them both below and in the manuscript.

We have provided point-by-point responses to the comments raised by the Reviewer below.

1. Line 33 "an" rather than "the"

Thank you for this observation. We have made this change.

2. Line 64 The increased density of nerve fibres referred to here is not obvious from the micrographs shown in Supplementary Fig 1A. The muscle layer is clearly thicker (although this could be made clearer by delineating the edges in some way), but density changes need quantification or higher magnification micrographs.

This is an important point and we have added higher magnification insets to Fig.1a.

3. Line 84 The work of Chevalier et al 2021 (Communications Biology) describing an equivalent grid in mouse ENS development is relevant here.

We agree that we missed an opportunity to cite this work and contextualize it within our own findings. We have added the citation and now compare our findings of a grid-like neuronal cell body network with Chevalier's characterization of the lattice formed by the enteric neuronal fibers.

In Results: *"The reorganization of neuron cell bodies generated stripes with an overlying grid-like structure, a structure reminiscent of but more elaborate than neuronal stripes in mouse. This observation of enteric neuron cell body organization complements the known structure of enteric neuronal processes, which form a lattice in mouse embryos."*

4. Line 91-94 This sentence over-simplifies the problem. Why was SOM used as a marker for interneurons?

Thank you for this question. We have added a more thorough explanation of marker selection in the body of the text and refer to Hamnett and Dershowitz et al., 2022, which delineates the cross-sectional approach we took comparing single-cell RNA sequencing (scRNAseq) and histology data to identify markers. In multiple scRNAseq screens, somatostatin (SST) is identified as a marker of putative interneuron populations.¹⁻³ Going through the ENS literature, we found evidence that SST antibodies label neuronal cell bodies in human and other species, which is essential for our quantification methods.^{4,5} Further, SST labels neuronal cell bodies in the adult human submucosal plexus.⁴ Thus, we determined that SST was an appropriate marker for labeling and quantifying interneurons in the developing human intestine.

In Results: “The basic GI motility circuit in rodents contains four functionally distinct neuron classes. scRNAseq studies of the mouse ENS have identified putative markers of enteric neuronal subtypes, and we have recently identified subsets of these markers that reliably label neuronal cell bodies from different enteric neuron classes in the adult mouse ENS. Using these prior studies as a basis, we next assessed the quantity and distribution of the main enteric neuron classes across human MP development.”

5. Line 132 what does “de-prioritization” mean? Delayed or failure of?

We agree that our usage of this term is confusing. “De-prioritization” was intended to mean that the intestine shows greater growth restriction than other organs. We have clarified this in the text.

In Discussion: “Understanding this structural progression is of major clinical significance for neonatology and maternal-fetal medicine because this process may be disrupted in cases of utero-placental insufficiency, which results in greater reduction of fetal intestine growth as compared to growth of other organs.”

6. Line 135-136 depends on whether calretinin is a universal or even a majority marker for excitatory motor neurons. What is the evidence about calretinin in human intrinsic sensory neurons or interneurons for example? Are there other excitatory motor neurons that are not calretinin?

This is an important distinction, and we thank the reviewer for this question. In mouse scRNAseq studies, calretinin (CalR) is a reliable marker for a large proportion of putative excitatory motor neurons.⁶ Per human scRNAseq, CalR labels a population of human fetal enteric neurons, yet whether these are putative excitatory motor neurons is unknown. We primarily selected CalR as a marker because it is clinically used by pathologists in assessment of the Hirschsprung’s intestine.⁷ We have changed the text to read that we are addressing CalR+ neurons.

In Results: “Thus, the mid-gestation human MP contains distinct neuron classes with apparent increasing representation of CalR+ subtypes and decreasing representation of nNOS+ subtypes.”

7. Line 142-143 The ripples described here are remarkably similar to those identified in prenatal and immediately postnatal mouse colon by Roberts et al 2007 and 2010. This seems an important parallel, especially as this is the last period during which both the human preterm infant and the developing mouse gut are both free of an internal microbiome with its clear effects on enteric neuronal maturation (Collins et al 2013, Hung et al 2019, Poon et al 2020).

We strongly agree with this point. We now reference these works and have contextualized our work in terms of what is known about mouse myogenic ripples.

In Discussion: “The timeline of GI motility maturation from myogenic ripples to neurogenic motility mirrors development of GI motility in mouse embryos, which exhibit myogenic ripples at mid-gestation and neurogenic GI motility just prior to birth.”

Reviewer #2:

In this manuscript, Dershowitz and colleagues use immunohistochemical staining and ex vivo motility assays to describe the neuroanatomy and contractility of the human fetal enteric nervous system (ENS) at 14-23 weeks post-conception (PCW). They describe their work as the “first description of human fetal ENS structural development” that is “foundational knowledge”. Their

major conclusions are that the ENS from mid-gestation fetuses undergo substantial reorganization, and that within the myenteric plexus (MP) there are distinct neuronal subtypes and representation of excitatory vs. inhibitory motor neuron populations that change over time. They found that starting at 21 PCW, contractions only propagated from proximal to distal. Using tetrodotoxin to block neuronal inputs, they found the ENS may first influence GI motility starting at 21PCW. The strengths of this manuscript are that it is clearly written, its focus on the human ENS, and examination of human fetal intestinal contractility. Many major deficits including the failure to acknowledge/cite previous work in this area, lack of experimental replicates, and superficial observations/characterization of the tissue, however, significantly diminish enthusiasm. These methodological issues limit what, if any, conclusions can be drawn from this work.

We appreciate Reviewer #2's critiques. We agree with the necessity to better contextualize our work, include more experimental replicates, as well as analyze non-neuronal tissues in addition to the ENS. To address these issues, we have expanded the Introduction to include a more thorough description of prior work, added new analyses of smooth muscle and glial cells, and also collected more samples to strengthen our observations.

We have incorporated Reviewer #2's suggestions and address them below.

Major concerns:

1. Previous groups have used IHC to describe enteric neurons and glia in relation to surrounding cells in the smooth muscle of the human fetal bowel, including in the second trimester. None of these studies were cited by the authors (PMID: 15672264, 32598482, 14991401, 12950074). These studies should be appropriately described and acknowledged to provide appropriate context for the current study, and also compel revision of the statement that the study under review is the "first description of human fetal ENS structural development".

We appreciate the reviewer highlighting these important oversights. We have taken two approaches to address them. First, we have added a paragraph to the introduction to better describe the literature on human fetal ENS development as well as development of non-neuronal intestinal tissues. In this paragraph, we have added all of the suggested citations including additional citations that describe the progressive organization of interstitial cells of Cajal over gestational time.

In Introduction: *"The ENS, an autonomous branch of the peripheral nervous system, resides within the intestinal walls and controls GI function. The ENS arises from migratory neural crest cells. In human these cells enter the developing intestine at 4PCW and have colonized the intestinal length by 7PCW, and this intrinsic innervation precedes extrinsic intestinal innervation, which is not present by 9PCW. Whole-intestine single-cell RNA sequencing (scRNAseq) has revealed that enteric neurons are present in the intestine as early as 6PCW and that genes indicative of functionally distinct enteric neuron classes are expressed during the first trimester. Visualization of the human fetal ENS using cross sections, which show the ENS within the intestinal walls, have identified structural changes with the emergence of individual ganglia evident at 14PCW."*

Further, we agree that our claim "first description of human fetal ENS structural development" is misleading, and we apologize for the overstatement. We have changed the text to clarify that we are providing the first description of human fetal ENS structure in wholemount preparations from the second trimester, which provide a different perspective from cross sections.

In Introduction: “Here, we provide the first assessment of human fetal ENS development **in wholemount** preparations from second trimester development, between 13-23PCW.”

2. Almost all of the data presented in Figures 1 and 2, and Supp Fig 1 have sample sizes of “1-3”. Human tissue is understandably difficult to acquire but showing data from a single subject and suggesting that it is “foundational knowledge” is inappropriate. Studies need to be powered for reproducibility. No reasonable biological conclusions can be drawn from observing motility in a single tissue sample, other than that contractions can be visualized *ex vivo* in the fetal intestine. Antenatal imaging already shows this to occur *in vivo*.

Furthermore, the way the data is presented in Figures 1f, 1i, 1j suggests that there are many samples from each PCW but the figure legend is not consistent. If there are multiple technical replicates from each biological replicate, then this should be clearly described as such.

We thank the reviewer for raising these important points.

We agree that sample size is a limitation of our study. We are grateful to Stanford Family Planning at Stanford School of Medicine that we were able to gather several more human samples during this revision period, and we now have n=4 samples with evidence of neurogenic GI motility. To clarify sample size further, we have included a table in the Methods section (Table 1) that outlines all samples we collected, which regions were collected, and how each sample was analyzed (see Table 1). We have also decided to pool fetal ages into early, mid, and late second trimester ages (13-15, 18-20, and 21-23 postconceptional weeks, respectively). This approach is routine in human fetal studies given the challenges of collecting these samples especially from the second trimester.^{1,8,9}

We appreciate the difficulty of calling work “foundational” with few replicates. We have taken care to thoughtfully revise the manuscript in order to more appropriately describe the impact and limitations of our work. We have removed words such as “foundational.” For example:

In Abstract: “*Here we demonstrate that the second trimester human fetal ENS takes on a striped organization akin to embryonic mouse. Further, we perform the first *ex vivo* functional assay of human fetal tissue and find that human fetal GI motility matures in a similar progression to embryonic mouse GI motility. Together, this provides critical knowledge, which facilitates comparisons with common animal models to advance translational disease investigations and testing of pharmacological agents to enhance GI motility in prematurity.*”

3. There are no data provided on the human subjects – inclusion/exclusion criteria, genetic defects, sex.

We thank the review for pointing out this important omission. As mentioned above, we have now included a table in the Methods section (Table 1) that describes all samples collected. Additionally, all submissions to Nature Journals require that information such as inclusion/exclusion criteria and sex be included in the mandatory “Reporting Summary” document, which is made readily available upon publication as part of the Supplementary Materials. Please see the Reporting Summary with the revised manuscript as well as the screenshots below:

Life sciences study design

All studies must disclose on these points even when the disclosure is negative.

Sample size	Samples were collected after elective pregnancy termination under a protocol approved through the Research Compliance Office at Stanford University. Given the sensitivity and rarity of this tissue, sample size was based on tissue availability.
Data exclusions	Samples were excluded from structural analysis if damage from the termination procedure and tissue collection disrupted our ability to identify individual enteric neurons. Regions of intestine found to have an incomplete myenteric plexus were excluded from motility analysis.
Replication	Each experiment included independent biological replicates if multiple samples of the same gestational age and intestinal region were available.
Randomization	Comparisons were made between regions of the intestines or between developmental time points, thus no randomization was necessary and groups were dictated by the nature of the sample.
Blinding	Given that our data were quantitative and not allocated into groups, blinding was not necessary for our analysis.

Human research participants

Policy information about studies involving human research participants and Sex and Gender in Research.

Reporting on sex and gender	Sex data was not available at the time of the collection of human fetal tissue.
Population characteristics	Samples consisted of fetal tissue collected from elective pregnancy terminations.
Recruitment	All women who consented to donate tissue for research were eligible to be included in our study. We collected samples from all non-disease carrying fetuses where we could clearly identify the intestine.
Ethics oversight	Per the Stanford Institute Tissue Donation Policy, no study specific consent is need for the use of donated tissue that is not going to be cultured or used to create a cell line.

4a. Lack of characterization of tissue: Conclusions are drawn about myogenic vs. ENS mediated contractions based on studying a single subject yet there is no characterization of the cells of the smooth muscle syncytium at these ages. No examination of enteric glia.

b. Similarly, they use calretinin and nNOS as markers of excitatory and inhibitory neurons, respectively, and draw conclusions about excitatory/inhibitory balance. However, it is not clear to what extent these designations are appropriate for the human fetal bowel. What is the evidence for this?

c. Finally, SST is a neuropeptide and IHC for most neuropeptides in the adult ENS does not identify cell soma well because the neuropeptides are trafficked to the processes. Thus, it is not clear that the quantification in Supp 1g. is meaningful. Supp Fig 1e clearly shows SST immunoreactivity in fibers.

Thank you for highlighting these points. We have addressed them each as described below:

a. We agree that the manuscript primarily focuses on neuronal cells rather than non-neuronal cells of the intestine. To address this, we have changed the title of the manuscript to “*Anatomical and functional maturation of the mid-gestation **human enteric nervous system**.*”

Additionally, we have collected more samples and now have n=5 samples that exhibit putative myogenic ripples based on their appearance and lack of response to tetrodotoxin, a characterization initially made in mouse embryos.^{10,11}

We also have sought to improve our characterization of non-neuronal tissues through histologically analyzing smooth muscle and glia. We have included quantifications of smooth muscle thickness from early and late in the second trimester (please see Supplementary Fig.1f,g). To address the question of glia, we have labeled samples from early and late in the second trimester with Sox10 (please see Supplementary Fig.1h,i), which is a marker of both glia and precursor cells.¹² In our characterizations of both smooth muscle and Sox10+ cells, we found

developmental differences between the small intestine and colon. This is an interesting distinction that we hope to address in future work.

b. Thank you for highlighting this question, which was also addressed by Reviewer #1. As above in response to Reviewer #1, question 6:

“This is an important distinction, and we thank the reviewer for this question. In mouse scRNAseq studies, calretinin (CalR) is a reliable marker for a large proportion of putative excitatory motor neurons.⁶ Per human scRNAseq, CalR labels a population of human fetal enteric neurons, yet whether these are putative excitatory motor neurons is unknown. We primarily selected CalR as a marker because it is clinically used by pathologists in assessment of the Hirschsprung’s intestine.⁷ We have changed the text to read that we are addressing CalR+ neurons.

In Results: “*Thus, the mid-gestation human MP contains distinct neuron classes with apparent increasing representation of CalR+ subtypes and decreasing representation of nNOS+ subtypes.*”

c. This is an interesting question that was also posed by Reviewer #1. As above in response to Reviewer #1, question 4:

“Thank you for this question. We have added a more thorough explanation of marker selection in the body of the text and refer to Hamnett and Dershowitz et al., 2022, which delineates the cross-sectional approach we took comparing single-cell RNA sequencing (scRNAseq) and histology data to identify markers. In multiple scRNAseq screens, somatostatin (SST) is identified as a marker of putative interneuron populations.¹⁻³ Going through the ENS literature, we found evidence that SST antibodies label neuronal cell bodies in human and other species, which is essential for our quantification methods.^{4,5} Further, SST labels neuronal cell bodies in the adult human submucosal plexus.⁴ Thus, we determined that SST was an appropriate marker for labeling and quantifying interneurons in the developing human intestine.

In Results: “*The basic GI motility circuit in rodents contains four functionally distinct neuron classes. scRNAseq studies of the mouse ENS have identified putative markers of enteric neuronal subtypes, and we have recently identified subsets of these markers that reliably label neuronal cell bodies from different enteric neuron classes in the adult mouse ENS. Using these prior studies as a basis, we next assessed the quantity and distribution of the main enteric neuron classes across human MP development.*”

Further, we agree that we see SST+ fibers, but we also see immunoreactivity in fibers for most of our markers (for example, nNOS in Fig.1j). However, the cell body labeling is still robust enough to allow for quantification of HuC/D+SST+ cells. To quantify neuronal cell bodies expressing a given marker, we first generate a mask with HuC/D, which labels all neuronal cell bodies (see Methods).¹³ This approach allows us to exclude fibers from our quantification.

5. There proximal to distal developmental gradient in the intestine is a well established concept for the ENS as well as many of the other key features of gut anatomy (epithelium, etc). Similarly, the stereotypic lattice-like structure of the ENS has been well described in numerous animal models and the human ENS over the course of many decades. Yet the authors re-brand this well-established organization as “neuronal stripes” and cite only their own study in mice as novel findings.

We agree that the lattice structure of enteric neuron fibers has been described in diverse organisms including human, mouse, and chick. In this revision, we have sought to better contextualize our work in the Results and Discussion sections within this well-established

framework of intestinal development. This point was also made by Reviewer #1. As above in response to Reviewer #1, question 3:

“We agree that we missed an opportunity to cite this work and contextualize it within our own findings. We have added the citation and now compare our findings of a grid-like neuronal cell body network with Chevalier’s characterization of the lattice formed by the enteric neuronal fibers.

In Results: *“The reorganization of neuron cell bodies generated stripes with an overlying grid-like structure, a structure reminiscent of but more elaborate than neuronal stripes in mouse. This observation of enteric neuron cell body organization complements the known structure of enteric neuronal processes, which form a lattice in mouse embryos.”*

Further, we developed a novel computational means to quantify the striped organization of neuronal cell bodies, and we have shown in both mouse and human that this method can be used to quantify patterning changes during ENS development (Fig. 1).¹³ Therefore, our work builds upon what is known about the lattice structure to add a new dimension of stripes and provide a means to quantify this patterning.

6. Cell death characterization is weak. This should either be quantified/reproduced or removed.

We thank the reviewer for this suggestion. We have added a quantification of Caspase3+HuC/D+ cells at early and late stages in the second trimester and found less than 0.6% of HuC/D+ neurons express the apoptotic marker Caspase3 (please see Supplementary Fig.1c-e). We have also included cross section images that demonstrate Caspase3 labeling in the epithelium, a tissue that undergoes significant apoptosis and serves as a positive control for our antibody labeling.

7. It is not clear what the analysis is in Fig 1e or what controls are done to illustrate that this is a meaningful assay for this tissue.

Thank you for this question. For the analysis in Fig.1e, we have utilized a conditional intensity function (CIF), which is a well-established algorithm for generating spatial probability maps and has been applied to many biological processes, most notably understanding firing patterns of large populations of neurons.¹⁴⁻¹⁶ We previously adapted this function to histologic data of enteric neuron cell bodies in embryonic and adult mouse, and we demonstrated how this analysis could be used to detect patterning differences both over time and by intestinal region.¹³ For more details on how we adapted this well-established paradigm to ENS histology data, please see Methods in Hamnett and Dershowitz et al., 2022.

8. Minor: Graphs in 2D/E seem to be missing 21 PCW, but text focuses on a change at 21PCW.

Thank you for pointing out this omission. We now include all ages in this graph and have also collected more samples to include in these data (please see Fig.2d).

Reviewer #3:

This is an interesting and necessary study by Dershowitz et al on the anatomical and functionality of mid gestation intestine. Some suggestions to improve the manuscript are provided below:

We greatly thank Reviewer 3 for their interest in our work and for their thoughtful comments. We have addressed them each below.

1. Lines 65-68 to summarize Figure 1 saying “develop substantially” in morphology are vague and descriptive and should be described in more detail. The same feedback for lines 87-88 (“substantial reorganization”).

Thank you for this suggestion. We agree that this language is vague. We have removed language such as “substantial.” Further, we have sought to more quantitatively describe changes in non-neuronal cells in the intestine and to use specific values to describe these changes (please see Supplementary Fig.1f-h).

2. For Figure 1C, 1D, 1E it is unclear what the pictures are showing with regard to the anatomic orientation of the intestine. An orientation picture would be helpful.

Thank you for suggesting that we add a schematic. We have added a diagram of the intestinal tube, cross section, and wholemount in Fig1.b, and we agree this makes the overall figure clearer.

3. For the data and results described in figure 2, how can the authors conclude that the ENS may first influence GI motility at 21 weeks without data from additional timepoints such as 15-20 weeks. In terms of the variability seen in humans, I would recommend these experiments be repeated with n= 3-4 per group at least, as an n of 1 is unacceptable to draw any conclusions.

During the revision process, we are grateful to Stanford Family Planning at Stanford School of Medicine to have collected more samples for functional GI motility analysis. We now have n=5 samples that exhibit only myogenic ripples (ages 13-15 postconceptional weeks) and n=4 samples that exhibit neurogenic GI motility (ages 18-22 postconceptional weeks). Please see Fig.2 and Supplementary Fig.2. As these samples are exceedingly rare, we believe that these data are sufficient to conclude that GI motility is limited to myogenic activity early in the second trimester and that neurogenic motility begins to mature later in the second trimester.

Minor comments for improvement are below:

4. Minimize the use of acronyms throughout as it makes it difficult to read the manuscript. Acronyms when used do need to be spelled out the first time.

Thank you for this suggestion. We have sought to reduce the number of abbreviations throughout the manuscript. For example, we now state “small intestine” rather than SI.

5. Videos didn't download correctly for this reviewer and so were unable to be viewed. Excessive use of videos is a weakness of the manuscript.

We apologize for the challenges viewing our videos. These are an important part of our manuscript, and we have taken three approaches to address this concern. First, we have added more examples of still images to Fig.2 in order to provide a visual alternative to videos. Second, we have reduced the number of supplementary videos. Finally, we have communicated with editors at *Nature Communications* to ensure that our videos are in the correct format and can be viewed by all potential readers.

References:

1. Fawkner-Corbett, D. *et al.* Spatiotemporal analysis of human intestinal development at single-cell resolution. *Cell* **184**, 810-826.e23 (2021).

2. Elmentaite, R. *et al.* Single-Cell Sequencing of Developing Human Gut Reveals Transcriptional Links to Childhood Crohn's Disease. *Dev. Cell* **55**, 771-783.e5 (2020).
3. Egozi, A. *et al.* Insulin is expressed by enteroendocrine cells during human fetal development. *Nat. Med.* 2021 2712 **27**, 2104–2107 (2021).
4. Kustermann, A., Neuhuber, W. & Brehmer, A. Calretinin and somatostatin immunoreactivities label different human submucosal neuron populations. *Anat. Rec. (Hoboken)*. **294**, 858–869 (2011).
5. Bulc, M., Palus, K. & Calka, J. The Influence of a Hyperglycemic Condition on the Population of Somatostatin Enteric Neurons in the Porcine Gastrointestinal Tract. *Anim. an open access J. from MDPI* **10**, (2020).
6. Morarach, K. *et al.* Diversification of molecularly defined myenteric neuron classes revealed by single-cell RNA sequencing. *Nat. Neurosci.* **24**, 34–46 (2021).
7. Cheng, L. S., Schwartz, D. M., Hotta, R., Graham, H. K. & Goldstein, A. M. Bowel dysfunction following pullthrough surgery is associated with an overabundance of nitrergic neurons in Hirschsprung disease HHS Public Access. *J Pediatr Surg* **51**, 1834–1838 (2016).
8. Fiock, K. L., Smalley, M. E., Crary, J. F., Pasca, A. M. & Hefti, M. M. Increased Tau Expression Correlates with Neuronal Maturation in the Developing Human Cerebral Cortex. *eNeuro* **7**, (2020).
9. Petrikin, J. E., Gaedigk, R., Leeder, J. S. & Truog, W. E. Selective Toll-like receptor expression in human fetal lung. *Pediatr. Res.* **68**, 335–338 (2010).
10. Roberts, R. R. *et al.* The first intestinal motility patterns in fetal mice are not mediated by neurons or interstitial cells of Cajal. *J. Physiol.* **588**, 1153–1169 (2010).
11. Roberts, R. R., Murphy, J. F., Young, H. M. & Bornstein, J. C. Development of colonic motility in the neonatal mouse—studies using spatiotemporal maps. *Am. J. Physiol. Liver Physiol.* **292**, G930–G938 (2007).
12. Boesmans, W., Lasrado, R., Vanden Berghe, P. & Pachnis, V. Heterogeneity and phenotypic plasticity of glial cells in the mammalian enteric nervous system. *Glia* **63**, 229–241 (2015).
13. Hamnett, R. *et al.* Regional cytoarchitecture of the adult and developing mouse enteric nervous system. *Curr. Biol.* 2021.07.16.452735 (2022) doi:10.1016/j.cub.2022.08.030.
14. Truccolo, W., Eden, U. T., Fellows, M. R., Donoghue, J. P. & Brown, E. N. A point process framework for relating neural spiking activity to spiking history, neural ensemble, and extrinsic covariate effects. *J. Neurophysiol.* **93**, 1074–1089 (2005).
15. Sarma, S. V. *et al.* Computing confidence intervals for point process models. *Neural Comput.* **23**, 2731–2745 (2011).
16. Tao, L., Weber, K. E., Arai, K. & Eden, U. T. A common goodness-of-fit framework for neural population models using marked point process time-rescaling. *J. Comput. Neurosci.* **45**, 147–162 (2018).

REVIEWERS' COMMENTS

Reviewer #1 (Remarks to the Author):

The authors have dealt with most of my concerns satisfactorily. The exception is in lines 127-130, where they make statements about specific immunohistochemical markers for different neuronal subtypes in mouse myenteric plexus that are much too sweeping. While the transcriptomic studies certainly speculate that calretinin is a marker for excitatory motor neurons and somatostatin is a marker for interneurons, there is abundant other evidence that calretinin can label some intrinsic sensory neurons and probably other interneurons, that NOS labels some interneurons in addition to inhibitory motor neurons and that somatostatin only labels a subset of interneurons. There is also evidence that NFM labels some neurons that are not intrinsic sensory neurons. See for example, Qu et al 2008, Sang & Young 1996. This problem can be addressed by inserting “subsets of” before “excitatory” and “a subset” before “interneurons” and acknowledging that the staining used in this study does not produce a comprehensive inventory of different functional subtypes.

Minor points

1 line 117 insert commas after “of” and “than”

2 line 190 the appearance of neurogenic motility in mouse is region dependent appearing in duodenum at E18.5, but being absent in colon before P6. This regional specificity should be acknowledged here.

Reviewer #2 (Remarks to the Author):

The manuscript by Dershowitz and colleagues is substantially improved by the revisions made. Specifically, the addition of more subjects to the motility studies and the more appropriate consideration/citation of the previous literature in this area are beneficial changes. Overall, the major advance of this manuscript is its focus on analysis of fetal ENS development in human tissues. This is an important area that has been understudied due to the constraints of sample acquisition. In this work, the authors study a small number of human fetal intestinal samples from PCW14-23 by whole mount immunohistochemistry for 5 markers: pan-neuronal marker HuC/D, Calretinin, NOS1, SST, and NF-M. They find that myenteric neurons become organized into the lattice-like ganglionated plexus network over this period, similar to what has been reported in mice. The proportions of neurons marked by the other antibodies may vary by age and intestinal region, but the magnitude and significance of any potential differences are difficult to assess the way the data are currently presented. Then, they examined intestinal contraction patterns *ex vivo* and find that a pattern of myogenic ripples becomes TTX-sensitive by 18PCW, suggesting that neurogenic motor patterns arise after myogenic ones, similar to mouse fetal development.

Overall, this work is important supportive evidence that many of the observations about ENS development made in rodent and chick models are consistent with what is likely occurring in humans. Given that this work is a largely qualitative study in many ways, because of the severe and understandable limitations of sample acquisition, there are some considerations that would help readers better interpret the findings presented.

1 – Information on human subjects – The inclusion of Table 1 in the Methods section is very helpful. From this and the reporting summary it still remains unclear if any of the subjects were had genetic defects. This is important information because major genetic defects like trisomies often prompt elective termination but could also affect ENS anatomy and function – does “non-diseased” fetuses mean that the subjects were confirmed not to have such defects?

2 – As raised previously, Figures 1g, 1i and 1k are very difficult to interpret and it is not clear what the criteria were for drawing trendlines and how these should be interpreted given that the sample size is listed as N = 1-8. Pooling subjects into three age groups is very reasonable, but it would be helpful to see the data for each tissue separately so that it is clear which analyses were conducted on 1 sample and which on 8.

3 – Lines 86-91: The conclusions about muscle thickness and density of non-gial cells changing cannot be supported by the extent of analyses provided. Two 12-micron sections were analyzed per each of 2 subjects. On any given cross-section of intestinal tissue, muscle thickness can vary this much even based on which part of the section the measurement is taken and how contracted a particular segment was when it was fixed. These statements should be revised or strengthened with additional data. Furthermore, the graphs in Ext Fig 1g and 1i indicate that 2 different subjects were analyzed per age but Methods Table 1 shows that colon was only collected from one individual in the 13-15 PCW stage – this should be reconciled. Accordingly, is data in Fig 1c only representative of a single subject since only proximal colon is shown?

4 – The authors cite the Morarach Nat Neuroscience study to support usage of calretinin (gene name Calb2) as defining excitatory motor neurons, but both this scSEQ work as well IHC studies that were previously published clearly show Calretinin transcript and protein expression in IPANs (sensory neurons) and interneurons so lines 129-130 should be revised. Another important scSEQ paper in the field (May-Zhang et al., Gastroenterology, 2021) also shows Calb2 as a marker of IPANs in both humans and mice.

Reviewer #3 (Remarks to the Author):

All of my comments were addressed.

REVIEWERS' COMMENTS

Reviewer #1 (Remarks to the Author):

The authors have dealt with most of my concerns satisfactorily. The exception is in lines 127-130, where they make statements about specific immunohistochemical markers for different neuronal subtypes in mouse myenteric plexus that are much too sweeping. While the transcriptomic studies certainly speculate that calretinin is a marker for excitatory motor neurons and somatostatin is a marker for interneurons, there is abundant other evidence that calretinin can label some intrinsic sensory neurons and probably other interneurons, that NOS labels some interneurons in addition to inhibitory motor neurons and that somatostatin only labels a subset of interneurons. There is also evidence that NFM labels some neurons that are not intrinsic sensory neurons. See for example, Qu et al 2008, Sang & Young 1996. This problem can be addressed by inserting “subsets of” before “excitatory” and “a subset” before “interneurons” and acknowledging that the staining used in this study does not produce a comprehensive inventory of different functional subtypes.

Minor points

1 line 117 insert commas after “of” and “than”

2 line 190 the appearance of neurogenic motility in mouse is region dependent appearing in duodenum at E18.5, but being absent in colon before P6. This regional specificity should be acknowledged here.

Thank you for these additional comments. For the major comment, we edited our language and included additional citations to acknowledge that our subtype markers do not fully encompass each neuronal class.

In Results: *We labeled MP wholemounts with antibodies against calretinin (CalR), neuronal nitric oxide synthase (nNOS), neurofilament-M (NF-M), and somatostatin (SST), which broadly define subsets of excitatory motor neurons, inhibitory motor neurons, sensory neurons, and interneurons in mice, respectively (Fig. 1h-k, Supplementary Fig. 1j-m).*¹⁻³ *Although these markers approximate these neuronal classes, some markers are expressed in other cell types such as CalR expression in sensory neurons and NF-M expression in motor neurons.*^{1,4,5}

For each minor point:

1. We have inserted these commas.
2. We have clarified regional differences in the onset of gastrointestinal motility in mouse in the introduction.

In Introduction: *These neuronal stripes arise from gradual reorganization of enteric neurons during late embryonic and early postnatal ages,*⁶ *which corresponds to the emergence of neurogenic GI motility at embryonic day 18.5 in the mouse small intestine.*^{7,8} *In mouse colon, the emergence of neuronal stripes and the onset of neurogenic GI motility both occur in the early postnatal period.*^{6,7}

Reviewer #2 (Remarks to the Author):

The manuscript by Dershowitz and colleagues is substantially improved by the revisions made. Specifically, the addition of more subjects to the motility studies and the more appropriate consideration/citation of the previous literature in this area are beneficial changes. Overall, the major advance of this manuscript is its focus on analysis of fetal ENS development in human tissues. This is an important area that has been understudied due to the constraints of sample acquisition. In this work, the authors study a small number of human fetal intestinal samples

from PCW14-23 by whole mount immunohistochemistry for 5 markers: pan-neuronal marker HuC/D, Calretinin, NOS1, SST, and NF-M. They find that myenteric neurons become organized into the lattice-like ganglionated plexus network over this period, similar to what has been reported in mice. The proportions of neurons marked by the other antibodies may vary by age and intestinal region, but the magnitude and significance of any potential differences are difficult to assess the way the data are currently presented. Then, they examined intestinal contraction patterns ex vivo and find that a pattern of myogenic ripples becomes TTX-sensitive by 18PCW, suggesting that neurogenic motor patterns arise after myogenic ones, similar to mouse fetal development.

Overall, this work is important supportive evidence that many of the observations about ENS development made in rodent and chick models are consistent with what is likely occurring in humans. Given that this work is a largely qualitative study in many ways, because of the severe and understandable limitations of sample acquisition, there are some considerations that would help readers better interpret the findings presented.

We are pleased that Reviewer 2 considers our revised manuscript to be significantly improved. We have addressed their comments in full below.

1 – Information on human subjects – The inclusion of Table 1 in the Methods section is very helpful. From this and the reporting summary it still remains unclear if any of the subjects were had genetic defects. This is important information because major genetic defects like trisomies often prompt elective termination but could also affect ENS anatomy and function – does “non-diseased” fetuses mean that the subjects were confirmed not to have such defects?

We are glad that Reviewer 2 found Table 1 helpful. We have elaborated in Methods about the collection of human fetal samples. We state that all collected tissues were negative for trisomies.

In Methods: *All tissues collected were negative for trisomies, based on prenatal screening. Tissue was not collected from pregnancies with prenatal screening positive for trisomies. All eligible, non-trisomy carrying samples were included in the study.*

2 – As raised previously, Figures 1g, 1i and 1k are very difficult to interpret and it is not clear what the criteria were for drawing trendlines and how these should be interpreted given that the sample size is listed as N = 1-8. Pooling subjects into three age groups is very reasonable, but it would be helpful to see the data for each tissue separately so that it is clear which analyses were conducted on 1 sample and which on 8.

Thank you for these comments. To address the difficulty with these graphs, we have included in the figure legends definitions of the significance of the trendlines, which connect the means of data for a given age. We have also more clearly defined the n for each age, region, and experiment within the accompanying figure legends.

For example, in Figure 1 Legend:

3 – Lines 86-91: The conclusions about muscle thickness and density of non-glial cells changing cannot be supported by the extent of analyses provided. Two 12-micron sections were analyzed per each of 2 subjects. On any given cross-section of intestinal tissue, muscle thickness can vary this much even based on which part of the section the measurement is taken and how contracted a particular segment was when it was fixed. These statements should be revised or

strengthened with additional data. Furthermore, the graphs in Ext Fig 1g and 1i indicate that 2 different subjects were analyzed per age but Methods Table 1 shows that colon was only collected from one individual in the 13-15 PCW stage – this should be reconciled. Accordingly, is data in Fig 1c only representative of a single subject since only proximal colon is shown?

We agree with the limitations of analyzing tissues in cross section, and we changed the language in the Results section to acknowledge these limitations. Also, thank the reviewer for catching the omission of samples in Table 1. We initially only included samples analyzed in wholemount. We have added samples analyzed only in cross section and denoted them with an * (Supplementary Table 1).

In Results: *We also observed morphological changes in other intestinal tissues, although this analysis is limited by sample number and potential variability in tissue embedding.*

4 – The authors cite the Morarach Nat Neuroscience study to support usage of calretinin (gene name Calb2) as defining excitatory motor neurons, but both this scSEQ work as well IHC studies that were previously published clearly show Calretinin transcript and protein expression in IPANs (sensory neurons) and interneurons so lines 129-130 should be revised. Another important scSEQ paper in the field (May-Zhang et al., Gastroenterology, 2021) also shows Calb2 as a marker of IPANs in both humans and mice.

Thank you for this comment, which is reminiscent of Reviewer 1's major comment. As written above: We edited our language and included additional citations to acknowledge that our subtype markers do not fully encompass each neuronal class.

In Results: *We labeled MP wholemounts with antibodies against calretinin (CalR), neuronal nitric oxide synthase (nNOS), neurofilament-M (NF-M), and somatostatin (SST), which broadly define subsets of excitatory motor neurons, inhibitory motor neurons, sensory neurons, and interneurons in mice, respectively (Fig.1h-k, Supplementary Fig.1j-m).¹⁻³ These markers approximate four neuronal classes, though CalR expression has also been observed in sensory neurons, and NF-M expression has been noted in some motor neurons.^{1,4,5}*

Reviewer #3 (Remarks to the Author):

All of my comments were addressed.

We thank Review 3 for helping us improve our manuscript. We are pleased to have addressed all comments.

References:

1. Morarach, K. *et al.* Diversification of molecularly defined myenteric neuron classes revealed by single-cell RNA sequencing. *Nat. Neurosci.* **24**, 34–46 (2021).
2. Fung, C. & Vanden Berghe, P. Functional circuits and signal processing in the enteric nervous system. *Cellular and Molecular Life Sciences* vol. 1 3 (2020).
3. Drokhlyansky, E. *et al.* The Human and Mouse Enteric Nervous System at Single-Cell Resolution. *Cell* **182**, 1606-1622.e23 (2020).
4. Qu, Z. D. *et al.* Immunohistochemical analysis of neuron types in the mouse small intestine. *Cell Tissue Res.* **334**, 147–161 (2008).
5. May-Zhang, A. A. *et al.* Combinatorial transcriptional profiling of mouse and human

- enteric neurons identifies shared and disparate subtypes in situ. *Gastroenterology* **160**, 755 (2021).
6. Hamnett, R. *et al.* Regional cytoarchitecture of the adult and developing mouse enteric nervous system. *Curr. Biol.* 2021.07.16.452735 (2022) doi:10.1016/j.cub.2022.08.030.
 7. Roberts, R. R., Murphy, J. F., Young, H. M. & Bornstein, J. C. Development of colonic motility in the neonatal mouse-studies using spatiotemporal maps. *Am. J. Physiol. Liver Physiol.* **292**, G930–G938 (2007).
 8. Roberts, R. R. *et al.* The first intestinal motility patterns in fetal mice are not mediated by neurons or interstitial cells of Cajal. *J. Physiol.* **588**, 1153–1169 (2010).